# Schistosoma japonicum leishmanolysin SjLLPi1 facilitates the invasion of cercariae into the host skin

Fanyan Chen[1,2]ᵒ, Bingkuan Zhu[3]ᵒ, Yu Fang[1]ᵒ, Zilüe Li[1]ᵒ, Zhigang Lei[1]ᵒ, Zechao Xue[1], Tao Shen[1], Sha Zhou[1], Xiaojun Chen[1], Lei Xu[1], Yalin Li[1], Jifeng Zhu[1]*, Wei Hu[3]*, Chuan Su[1]*

**1** Key Laboratory for Pathogen Infection and Control of Jiangsu Province, Department of Pathogen Biology and Immunology, School of Basic Medical Sciences, National Vaccine Innovation Platform, Nanjing Medical University, Nanjing, Jiangsu, China, **2** Department of Laboratory Medicine, The Eighth Hospital of Wuhan, Wuhan, Hubei, China, **3** State Key Laboratory of Genetic Engineering, Ministry of Education Key Laboratory of Contemporary Anthropology, School of Life Sciences, Fudan University, Shanghai, China

ᵒ These authors contributed equally to the work.

* chuansu@njmu.edu.cn (CS); huw@fudan.edu.cn (WH); zhujifengfinn@njmu.edu.cn (JZ)

## Abstract

### Background

Schistosomiasis is an important neglected tropical disease necessitating focus. Cercarial proteases are essential for schistosome invasion. Leishmanolysin has been identified as the most predominant protease in *Schistosoma japonicum* (*S. japonicum*) cercariae, but the role and mechanism of leishmanolysin in host skin invasion by *S. japonicum* cercariae remain unclear.

### Methodology/principal findings

Our bioinformatic analysis revealed the classification of *S. japonicum* leishmanolysin within the M8 matrix metalloprotease family. We then expressed recombinant *S. japonicum* leishmanolysin-like peptidase isoform 1 (SjLLPi1) and verified its hydrolytic enzyme activity. Western blotting analysis confirmed high level of SjLLPi1 protein in *S. japonicum* cercariae. Immunofluorescence staining revealed SjLLPi1 is predominantly present in the acetabular glands and their ducts in the cercarial head. Infection of mice with anti-SjLLPi1 monoclonal antibody treated *S. japonicum* cercariae significantly reduced worm and egg burden in mice 42 days post-infection. Infection of mice with anti-SjLLPi1 monoclonal antibody treated *S. japonicum* cercariae also significantly reduced parasite number in mice 7 days post-infection. In addition, treatment of mouse macrophages with SjLLPi1 prompted notable macrophage activation and substantial parasiticidal NO release. Finally, mice infected with anti-SjLLPi1 monoclonal antibody treated cercariae demonstrated a marked reduction in skin-invading parasite numbers as early as 30 min post-infection.

**Data availability statement:** The authors confirm that the data supporting the findings of this study are available within the article and its Supporting information.

**Funding:** This work was supported by the grant of National Key R&D Program of China (https://service.most.gov.cn/; 2024YFC2309701 and 2024YFC2309703 to CS) and the grant of the National Natural Science Foundation of China (https://www.nsfc.gov.cn/; NSFC No: 82030061 to CS). The funders had no role in study design, data collection and analysis, decision to publish, or preparation of the manuscript.

**Competing interests:** The authors have declared that no competing interests exist.

## Conclusions/significance

Our study indicates that SjLLPi1 aids *S. japonicum* cercariae penetration into the definitive host by hydrolyzing skin components, thereby facilitating parasite migration and transition to adult worms within the host. These results may provide valuable guidance for vaccine development and control strategy formulation against schistosome infection.

## Author summary

Schistosomiasis, affecting over 200 million individuals globally, remains a significant neglected tropical disease. The cercaria stage of schistosomes is the only phase capable of infecting humans, with proteases from the acetabular glands of cercariae aiding in skin invasion. Leishmanolysin, which is the most predominant protease expressed by *S. japonicum* cercariae, has an unclear role in human skin invasion by *S. japonicum* cercariae. Bioinformatic analysis revealed that *S. japonicum* leishmanolysin belongs to the M8 matrix metalloprotease family, potentially with hydrolytic activity. We expressed recombinant *S. japonicum* leishmanolysin-like peptidase isoform 1 (SjLLPi1) and confirmed its enzymatic activity. Then we prepared the anti-SjLLPi1 monoclonal antibody. Treatment of *S. japonicum* cercariae with anti-SjLLPi1 monoclonal antibody reduced worm and egg burden in mice by preventing skin invasion, offering insights for vaccine development and control strategies.

## Introduction

Schistosomiasis, an extensive, insidious, and severe tropical parasitic disease affecting 78 countries, impacted an estimated 251 million individuals requiring treatment in 2021 [1,2]. This disease is primarily caused by three main species of schistosomes: *Schistosoma mansoni* (*S. mansoni*), prevalent in Africa and South America; *Schistosoma japonicum* (*S. japonicum*), present in Asia; and *Schistosoma haematobium* (*S. haematobium*) in Africa and the Middle East [3–5]. The life cycle of schistosomes involves multiple developmental stages, with cercariae being the exclusive stage capable of infecting definitive host humans and mammals [6,7]. When humans or mammals come into contact with water containing schistosome cercariae, these larvae can rapidly penetrate the skin of the definitive host, initiating an infection [8]. Therefore, hindering the invasion of cercariae into the definitive host represents a critical measure in the prevention of schistosome infections and the control of its transmission.

In the course of invading the definitive host, cercariae need to traverse the epidermis, basement membrane, and dermis [9,10]. This process necessitates the coordinated interplay of the sucker, acetabular glands, and tail movements. The acetabular gland plays a pivotal role during skin penetration by cercariae [11]. Cercariae are

equipped with two pairs of pre-acetabular glands and three pairs of post-acetabular glands located in the head. These acetabular glands facilitate invasion through the release of mucus and proteases as cercariae penetrate the skin [12]. Notably, the hydrolytic activity of proteases leads to the degradation of large molecules in skin [10,13], which plays a critical role in the breakdown of the epidermis and dermis, ultimately determines the efficacy of cercarial skin penetration [14]. Therefore, finding essential components in acetabular gland proteases that significantly impact cercarial penetration is crucial for impeding cercarial invasion and controlling the prevalence of schistosomiasis.

During the process of cercarial penetration of the host skin, proteases from various families are involved [15]. Research on *S. mansoni* indicates that the serine protease elastase plays a critical role in the invasion of cercariae into the host skin [16,17]. Elastase is responsible for breaking down macromolecular proteins in the host epidermis and dermis, such as keratin, elastin, collagen, fibronectin, and laminin, and also disrupting connections between the cells within the host epidermis [18]. It is noteworthy that *S. japonicum* cercariae penetrate the host skin much faster than *S. mansoni* cercariae, leading to speculation that the composition of proteases, either in type or quantity, utilized by *S. japonicum* cercariae during invasion may differ from that of *S. mansoni* cercariae [19,20].

Our previous study on the proteomics analysis of *S. japonicum* cercariae revealed that the leishmanolysin, a member of the matrix metalloprotease family, is the most abundant among the various proteases expressed by the cercariae [21]. However, the specific function of this protease during the cercarial stage remains unclear. Leishmanolysin, which belongs to the M8 matrix metalloprotease family, possesses the conserved zinc-binding motif (HEXXH) of matrix metalloproteases [22,23]. It exhibits enzymatic activity towards peptide substrates and can hydrolyze a range of compounds including casein, gelatin, albumin, hemoglobin, and fibrinogen [24,25]. Therefore, we postulate that the presence of leishmanolysin in *S. japonicum* cercariae may facilitate their penetration of host skin. Experimental validation is needed to confirm this hypothesis.

In this study, we revealed the involvement of SjLLPi1 from the acetabular glands in facilitating host skin invasion of *S. japonicum* cercariae by breaking down proteins in the host skin. Our findings enhance the understanding of the mechanisms involved in the invasion of *S. japonicum* cercariae, providing valuable insights that could contribute to the prevention and control of the prevalence of schistosomiasis japonica.

## Materials and methods

### Ethics statement

The animal studies were carried out in strict accordance with the recommendations in the Guide for the Care and Use of Laboratory Animals of the Ministry of Science and Technology of the People's Republic of China. The animal protocol was approved by Institutional Animal Care and Use Committee (IACUC) of Nanjing Medical University (Approval Number: IACUC-JIPD-IACUC 2110001).

### Bioinformatics analysis

Homologous sequence alignment was performed using NCBI Protein BLAST (https://blast.ncbi.nlm.nih.gov/Blast.cgi). Protein sequence alignment was conducted with ClustalX. The molecular weight and isoelectric point of SjLLPi1 were predicted using the Compute pI/MW tool on the Expert Protein Analysis System (ExPASy) website (https://www.expasy.org/). The SMART tool was utilized to analyze the structural domains of SjLLPi1 (http://smart.embl-heidelberg.de/smart). Protein signal peptide was predicted using SignalP (http://www.cbs.dtu.dk/services/SignalP). The secondary structure of SjLLPi1 was predicted using the Self-Optimized Prediction Method with Alignment (SOPMA, https://npsa-prabi.ibcp.fr/cgi-bin/npsa_automat.pl?page=npsa_sopma.htm). Multiple protein sequence alignment was carried out using ESPript (http://espript.ibcp.fr/ESPript/cgi-bin/ESPript.cgi). A phylogenetic tree was constructed using the maximum likelihood method by MEGA 11 software with leishmanolysins from *S. japonicum* (TNN18159.1), *S. mansoni* (XP_018646419.1), *S.*

*haematobium* (XP_035588688.2), *Schistosoma margrebowiei* (CAH8586321.1), *Trichobilharzia regenti* (CAH8853212.1), *Fasciola hepatica* (THD21211.1), *Fasciola gigantica* (TPP62261.1).

A spatial model of SjLLPi1 was constructed by homology modeling as described previously [26]. Briefly, The X-ray structure of *Leishmania major* leishmonolysin (PDB code: 1IMI.1A) was used as the template. The homology model was generated by SWISS-MODEL using the Basic Modeling method. The model was then refined using Chiron (https://dokhlab.med.psu.edu/chiron/login.php) and evaluated with SAVES v5.0 (http://servicesn.mbi.ucla.edu/SAVES/). The docking between SjLLPi1 and inhibitor 1,10-phenanthroline was performed using DiscoveryStudio (DS) 2022 software (BIOVIA, San Diego, CA). Molecular images were generated using PyMOL Molecular Graphics System (Schrödinger, Palo Alto, CA).

## Production of recombinant SjLLPi1 protein

The expression and purification of recombinant SjLLPi1 were conducted as previously described [26]. The open reading frame (ORF) of SjLLPi1 with a C-terminal $His_6$ tag was custom-synthesized by GenScript (Piscataway, NJ) and integrated into the pPIC9k vector. The pPIC9K-SjLLPi1 plasmid, linearized with *Sal*I, was utilized to transform the methylotrophic yeast *Pichia pastoris* GS115 strain via electroporation with a GenePulser Xcell (BioRad Laboratories, Hercules, CA). Positive recombinant *P. pastoris* clone was cultured in 100 ml BMGY medium until an $OD_{600}$ of 3–4 was reached. Subsequently, the cells were induced for protein expression with 0.5% methanol (v/v) every 24 hours. After a 5-day induction, the medium was collected by centrifugation at 13,000 g for 10 min. The supernatant containing SjLLPi1 was purified using $Ni^{2+}$-NTA affinity chromatography (GE Healthcare, Chicago, IL) followed by DEAE affinity chromatography (GE Healthcare). The peak fractions were pooled and concentrated using an Amicon Ultra-10K device (Merck Millipore, Billerica, MA), dialyzed against 1xPBS (pH 7.4), and quantified using a Pierce BCA Protein Assay (Thermo Fisher Scientific, Waltham, MA). Purity was evaluated by SDS-PAGE on 4–20% polyacrylamide SurePAGE Bis-Tris gels (GenScript) followed by Coomassie staining. The purified SjLLPi1 protein was aliquoted and stored at -80°C.

## Preparation of anti-SjLLPi1 monoclonal antibody

Monoclonal antibody against SjLLPi1 was produced using hybridoma technology by Shanghai Youke Biotechnology Co., Ltd (Shanghai, China). Female BALB/c mice were intravenously immunized with recombinant SjLLPi1 mixed with Freund's complete adjuvant (Sigma-Aldrich, Shanghai, China). Subsequent boosts were administered at a two-week interval using a mixture of recombinant SjLLPi1 and Freund's incomplete adjuvant (Sigma-Aldrich). Mice were euthanized seven days after the final injection, and spleens were collected for hybridoma cell line generation as done previously [27]. Positive hybridomas were subcloned four times after screening. Ascites fluid was obtained from liquid paraffin-primed BALB/c mice through intraperitoneal injection of $1.0x10^6$ hybridoma cells, and harvested 7–10 days post-injection. IgG was purified from the ascitic fluid using a protein G column (Thermo Fisher Scientific) following the manufacturer's guidelines.

## Collection of parasites at different developmental stages

**Miracidia collection.** Following the collection of eggs from the liver of *S. japonicum* infected mice, the eggs were transferred to dechlorinated water and exposed to light at 28°C for 6 hours. After allowing the suspension to settle on ice for 2 hours, centrifuged at 8000 × g for 1 min, the pellet was washed with PBS, and then centrifuged for collection.

**Sporocysts collection.** After infecting *Oncomelania hupensis* with miracidia for 5–6 weeks, the snails were crushed, and sporocysts were selected from the snail fragments under a microscope.

**Cercariae collection.** *Oncomelania hupensis* infected with miracidia for 7–8 weeks were collected and exposed to light at 28°C in dechlorinated water for 3 hours. For the preparation of cercarial secretions and cercarial sonicated supernatants, cercariae were collected using a capillary pipette into pre-chilled RPMI 1640 or PBS. For real-time

RT-PCR and western blotting analyses, dechlorinated water containing cercariae was left to settle on ice for 2 hours, then centrifuged at 8000 × g for 5 min, the pellet was washed with PBS and centrifuged for collection.

**Schistosomula collection.** After infecting mice with 50 *S. japonicum* cercariae for 30 min, the infected skin area was excised and cut into small fragments, the fragments were collected in Hank's solution. Subsequent to a 3-hour incubation at 37°C, the mixture was centrifuged at 500 × g for 5 min and rinsed with PBS, then underwent further centrifugation to collect the parasites.

7 days post-infection with 100 *S. japonicum* cercariae, mice were euthanized and perfused with PBS to collect the liquid flushed out from the portal vein. Following at 500 × g for 5 min, the pellet was resuspended in PBS for parasite counting.

## Adult worm collection

6 weeks post-infection with 100 *S. japonicum* cercariae, mice were euthanized, perfused with PBS, and adult worms flushed out were collected from the portal vein.

## Preparation of cercarial secretions and cercarial sonicated supernatant

**Preparation of cercarial secretions.** 10,000 cercariae collected from *Oncomelania hupensis* were placed in pre-chilled RPMI 1640 and vortexed for 1 min to induce tail loss, followed by incubation at 37 °C with 5% $CO_2$ for 3 h. The culture was centrifuged at 130 × g for 5 min, and the supernatant was collected, supplemented with protease inhibitor (EpiZyme, Shanghai, China), and lyophilized.

**Preparation of cercarial sonicated supernatant.** 10,000 cercariae collected from *Oncomelania hupensis* were suspended in PBS with protease inhibitor (EpiZyme), sonicated, and centrifuged at 13800 × g for 15 min.

The lyophilized samples reconstituted in PBS and sonicated supernatants were used for enzymatic activity assay.

## Enzymatic activity assay of recombinant SjLLPi1 protein, cercarial secretions and cercarial sonicated supernatant

The enzymatic activity of recombinant SjLLPi1 was evaluated using the fluorogenic substrate Suc-Leu-Tyr-AMC (Medbio, Shanghai, China). The recombinant SjLLPi1 protein (50 nM) was pre-incubated at 37 °C for 10 min in 100 µl buffer containing 50 mM Tris-HCl, 5 mM $CaCl_2$, and 1 µM $ZnCl_2$ at pH 7.4. Then the substrate (100 µl; 10 µM) in the same buffer was added, and the reaction was allowed to proceed for 10 min. A control reaction was conducted with buffer instead of the enzyme. The fluorescence release at excitation and emission wavelengths of 360 nm and 460 nm was measured using a Microplate Reader (BioTek, Winooski, VT)

The dose-dependent enzymatic activity of recombinant SjLLPi1 was assessed using an MMP Activity Assay Kit (Abcam, Cambridge, MA) following the manufacturer's instructions at concentrations of 0, 25, 50, 100, and 200 nM. The same kit was used to evaluate the enzymatic activities of cercarial secretions and cercarial sonicated supernatants, following the manufacturer's instructions. For antibody-treated groups, 0.5 mg/ml of either anti-SjLLPi1 antibody or control mouse IgG was used.

## Proteolytic activity assay of SjLLPi1 against host skin proteins

Recombinant SjLLPi1 (2 µM) was mixed 1:1 with 2 mM 4-Aminophenylmercuric Acetate (APMA, from the MMP Activity Assay Kit, Abcam) and pre-incubated at 37 °C for 1 h. An equal volume of buffer (also from the MMP Activity Assay Kit, Abcam) containing 20 µg of human keratin K1 (TargetMol, Shanghai, China), human keratin K10 (TargetMol), human type IV collagen (Aladdin, Shanghai, China), or human type I collagen (Sigma-Aldrich) was then added as substrate and

incubated at 37 °C for 48 h. The reaction products were then subjected to SDS-PAGE and stained with Coomassie brilliant blue R250 (Sigma-Aldrich) to assess substrate hydrolysis.

## SDS-PAGE

The samples were separated by SDS-PAGE (15% w/v separating gel) employing a Mini-PROTEAN Tetra Electrophoresis System (Bio-Rad Laboratories). Protein calibration markers from Thermo Fisher Scientific were utilized as size standards. Visualization of proteins was achieved by staining with Coomassie brilliant blue R250 (Sigma-Aldrich).

## Western blotting

Protein fractions resolved via SDS-PAGE were transferred onto a polyvinylidene difluoride membrane (Whatman Inc., Florham Park, NJ). Subsequently, the membrane was blocked with 5% skimmed milk and probed with anti-SjLLPi1 or anti-rabbit/human β-actin (Cell Signaling Technology, Danvers, MA) antibodies. Visualization of the blots was achieved using the Pierce ECL Plus Western Blotting Substrate (Thermo Fisher Scientific) and detected utilizing the ChemiDoc Touch Imaging System (BioRad Laboratories).

## Immunofluorescence microscopy

Immunofluorescence analysis was performed as previously described [28,29], with some modifications. Cercariae were fixed in 4% paraformaldehyde for 25 min at room temperature, washed in PBSTw (PBS + 0.1% Tween-20), and permeabilized at 37 °C for 40 min in 2 µg/ml Proteinase K and 0.5% SDS in PBSTw. Then samples were post-fixed for 10 min. After permeabilization, samples were washed in PBSTs and blocked (5% horse serum, 0.45% fish gelatin, 0.3% Triton X-100, 0.05% Tween-20 in PBS) at room temperature for 4 h. After blocking, samples were incubated overnight at 4 °C with anti-SjLLPi1 antibody. After >6 h washing, Alexa Fluor 488-conjugated goat anti-mouse IgG (Abcam) was applied overnight at 4 °C. DAPI was included in the final wash (>6 h). Samples were mounted with Antifade Mounting Medium (YEASEN, Shanghai, China) and detected using a Nikon positive confocal laser-scanning microscope using 40 × immersion objective. Controls included pre-immune IgG and a no-primary antibody control.

## Mice infection

*Oncomelania hupensis* containing *S. japonicum* cercariae (Chinese mainland strain) were obtained from the Parasitic Disease Prevention and Research Institute of Jiangsu Province. *Oncomelania hupensis* were placed in dechlorinated water and exposed to light for 3 hours to stimulate the shedding of cercariae. Cercariae were collected using a capillary pipette and transferred onto a coverslip. Under a stereomicroscope, 50 or 100 cercariae was counted. Anti-SjLLPi1 monoclonal antibody or control mouse IgG or control monoclonal antibody against schistosomal heat shock protein 60 (anti-HSP60) [30] was added to the cercaria-containing water at a final concentration of 0.5 mg/ml. The mixture was incubated for 10 minutes at room temperature prior to infection. Eight-week-old male C57BL/6 mice (purchased from Animal Experiment Center of Nanjing Medical University) were lightly anesthetized with isoflurane (2% for induction, 1% for maintenance). Abdominal fur was removed using an electric pet shaver to expose the skin, which was then moistened with dechlorinated water. The coverslip containing the cercariae was then applied to the abdominal skin of the mouse. After 15-20 min, the coverslip was removed, completing the infection procedure..

## Quantification of liver, intestinal, and fecal eggs

Mice were sacrificed 42 days post-infection by cervical dislocation under deep anesthesia with 5% isoflurane, and the whole liver and small intestine were collected post mortem, washed with pre-chilled PBS, dried, and weighed (denoted as

W). The liver and small intestine were then cut into fragments and digested at 37°C in 20 ml of 5% KOH for 2 hours. The suspension was centrifuged at 1700×g for 5 min, the pellet was washed, re-suspended in 1 ml PBS for egg counting.

Fecal samples from *S. japonicum* infected mice at 6 weeks post-infection were collected, weighed (denoted as W), dissolved in PBS, filtered through an 80-mesh filter, centrifuged at 1700×g for 5 min, and the pellet were re-suspended in 1 ml PBS for egg counting.

A 10 µl aliquot of the suspension was placed on a microscope slide, and all eggs within the 10 µl suspension were counted under a microscope (denoted as n). The number (denoted as N) of eggs per gram of liver, intestine, or feces was calculated using the following formula: $N = n \times 100/W$.

## Generation and treatment of bone marrow-derived macrophages (BMDM)

Bone marrow was collected from the femurs and tibias of eight-week-old male C57BL/6 mice, and homogenized by pipetting. After centrifugation, cells were incubated with red blood cell (RBC) lysis buffer (10 mM $KHCO_3$, 155.2 mM $NH_4Cl$, 99.4 µM EDTA-2Na) at 4°C for 3 minutes to remove RBCs. Then cells were resuspended in DMEM containing 10% FBS and 1% penicillin-streptomycin. Cells were seeded at a density of $2 \times 10^6$ cells per well in a 12-well plate, supplemented with 1.5 ml of DMEM containing 20 ng/ml M-CSF (MedChemExpress, Shanghai, China), 10% FBS, and 1% penicillin-streptomycin, and cultured at 37°C with 5% $CO_2$ for 7 days to obtain BMDM. Obtained BMDMs were resuspended in PBS containing 1% FBS and incubated with FITC-conjugated anti-mouse F4/80 antibody (Thermo Fisher Scientific) and APC-conjugated anti-mouse CD11b antibody (Thermo Fisher Scientific) at 4°C for 30 minutes. After staining, cells were analyzed by flow cytometry to determine the proportion of $F4/80^+CD11b^+$ BMDMs, which was used to assess the purity of the isolated macrophage population. For the treatment of BMDM with SjLLPi1, the cells were exposed to 20 µg/ml of recombinant SjLLPi1 for 24 hours.

## RNA isolation and real-time reverse transcription–polymerase chain reaction (RT-PCR)

Total RNA was isolated utilizing TRIzol reagent (Thermo Fisher Scientific), and the concentration and purity of RNA were assessed employing a NanoDrop 1000 spectrophotometer (Thermo Fisher Scientific). Subsequent to manufacturer guidelines, reverse transcription was performed utilizing an mRNA reverse transcription kit (Thermo Fisher Scientific).

Real-time RT-PCR was carried out utilizing SYBR Green Master Mix (Applied Biosystems, Foster City, CA). Fold changes in mRNA expression were calculated by the $2^{-\Delta\Delta Ct}$ method.

The following primers were used:

mouse *β-actin*, forward: GGCTGTATTCCCCTCCATCG, reverse: CCAGTTGGTAACAATGCCATGT; mouse *SjLLPi1*, forward: GGAAGTTGTCATCAGTAT, reverse: ACCATCAGTTACATTGAA; mouse *TNF-α*, forward: CCCTCACACTCAGATCATCTTC, reverse: GCTACGACGTGGGCTACAG; mouse *iNOS*, forward: GCCACCAACAATGGCAACA, reverse: CGTACCGGATGAGCTGTGAATT; mouse *IL-6*, forward: GAGGATACCACTCCCAACAGACC, reverse: AAGTGCATCATCGTTGTTCATACA; mouse *IL-10*, forward: ACTTTAAGGGTTACTTGGGTTGC, reverse: ATTTTCACAGGGGAGAAATCG; mouse *Arg-1*, forward: CAGAAGAATGGAAGAGTCAG, reverse: CAGATATGCAGGGAGTCACC.

## Nitric oxide detection

A Total Nitric Oxide Assay Kit (Beyotime, Shanghai, China) was used to determine the levels of NO in cell supernatant according to the manufacturer's instructions.

## Statistical analysis

GraphPad prism software 5.0 (GraphPad Prism Inc., San Diego, CA) was used for all statistical analysis using one-way ANOVA. $p \leq 0.05$ was considered statistically significant. P values are as indicated by asterisks: *$P \leq 0.05$, **$P \leq 0.01$, ***$P \leq 0.001$.

## Results

### SjLLPi1 belongs to the M8 matrix metalloprotease family

Firstly, we obtained the sequence of SjLLPi1 protein from the NCBI Protein database (accession number: TNN18159.1). The molecular weight of SjLLPi1 was estimated to be 67.67 kDa with an isoelectric point of 9.03 using Compute pI/MW tool on ExPASy website. Subsequently, the secondary structure of SjLLPi1 was predicted by SOPMA to consist of 23.14% α-helix, 17.40% β-strand, 4.56% β-turn, and 54.90% random coil (Table 1). Further analysis using the SMART tool unveiled that SjLLPi1 harbors two domains: a signal peptide at the N-terminus (residues 1–21) and a peptidase M8 domain (residues 114–592) (Fig 1A).

To explore the homology of SjLLPi1 with leishmanolysin from different schistosomes, sequence alignment was performed with leishmanolysins from various schistosome species using Protein BLAST. The results showed that SjLLPi1 shares 36% and 64% homology with leishmanolysins from *S. mansoni* and *S. haematobium*, respectively. Phylogenetic analysis indicated that SjLLPi1 and leishmanolysin from *S. haematobium* were clustered into the same branch (Fig 1B), implying a strong homologous relationship between them. Results of multiple sequence alignment showed that SjLLPi1 has the conserved $Zn^{2+}$ binding motif HEXXH and H and M residues on the distal end (Fig 1C), which are characteristics of the M8 matrix metalloprotease family [22,23], indicating SjLLPi1 belongs to the M8 matrix metalloprotease family.

Given that the enzymatic function of M8 matrix metalloproteases typically needs $Zn^{2+}$ for catalysis [31], to further confirm the M8 matrix metalloprotease activity of SjLLPi1, we conducted homology modeling based on the amino acid sequence of SjLLPi1 (template: 1 lmI.1.A, leishmanolysin). The results unveiled that two histidine residues and one glutamate residue within the structure of SjLLPi1 can interact with $Zn^{2+}$ (Fig 1D). Furthermore, a molecular interaction assay was conducted on SjLLPi1 in conjunction with the well-known M8 matrix metalloprotease inhibitor 1,10-phenanthroline [32,33]. The results showed that 1,10-phenanthroline could be docked into the active site pocket of SjLLPi1 (Fig 1E) and bind to the active site (Fig 1F). These observations indicate the potential of SjLLPi1 to exhibit enzymatic function associated with M8 matrix metalloproteases.

### Expression, purification, and enzymatic activity identification of recombinant SjLLPi1

Due to the presence of a signal peptide at the N-terminus of the SjLLPi1 sequence and the secretion capability of the *Pichia pastoris* expression system, we chose to express a portion of the SjLLPi1 protein sequence devoid of the signal peptide, specifically amino acids 22–592 (Fig 2A). We first constructed a recombinant plasmid pPIC9K containing the SjLLPi1 gene, which was then transformed into *Pichia pastoris* for expression. The purified protein was subjected to SDS-PAGE analysis, followed by Coomassie brilliant blue staining, results revealed a protein with molecular weight close to 66.2 kDa (Fig 2B), consistent with the theoretical molecular weight 65.15 kDa.

To further confirm that the purified protein was indeed SjLLPi1, we started to generate monoclonal antibody against SjLLPi1. After analyzing the hydrophobicity, homology, antigenicity, and hydrophilicity of SjLLPi1, we selected amino acids 303–592 from the SjLLPi1 protein sequence as the antigen for preparing anti-SjLLPi1 monoclonal antibody (Fig 2A). The antibodies produced were used to detect the purified SjLLPi1 protein, and the results revealed a specific band between 55 kDa and 70 kDa (Fig 2C), confirming that the purified protein was SjLLPi1.

**Table 1. Prediction of the secondary structure of SjLLPi1.**

|  | α-helix | β-strand | β-turn | Random coil |
|---|---|---|---|---|
| Rate | 23.14% | 17.40% | 4.56% | 54.90% |
| Number | 137 | 103 | 27 | 325 |

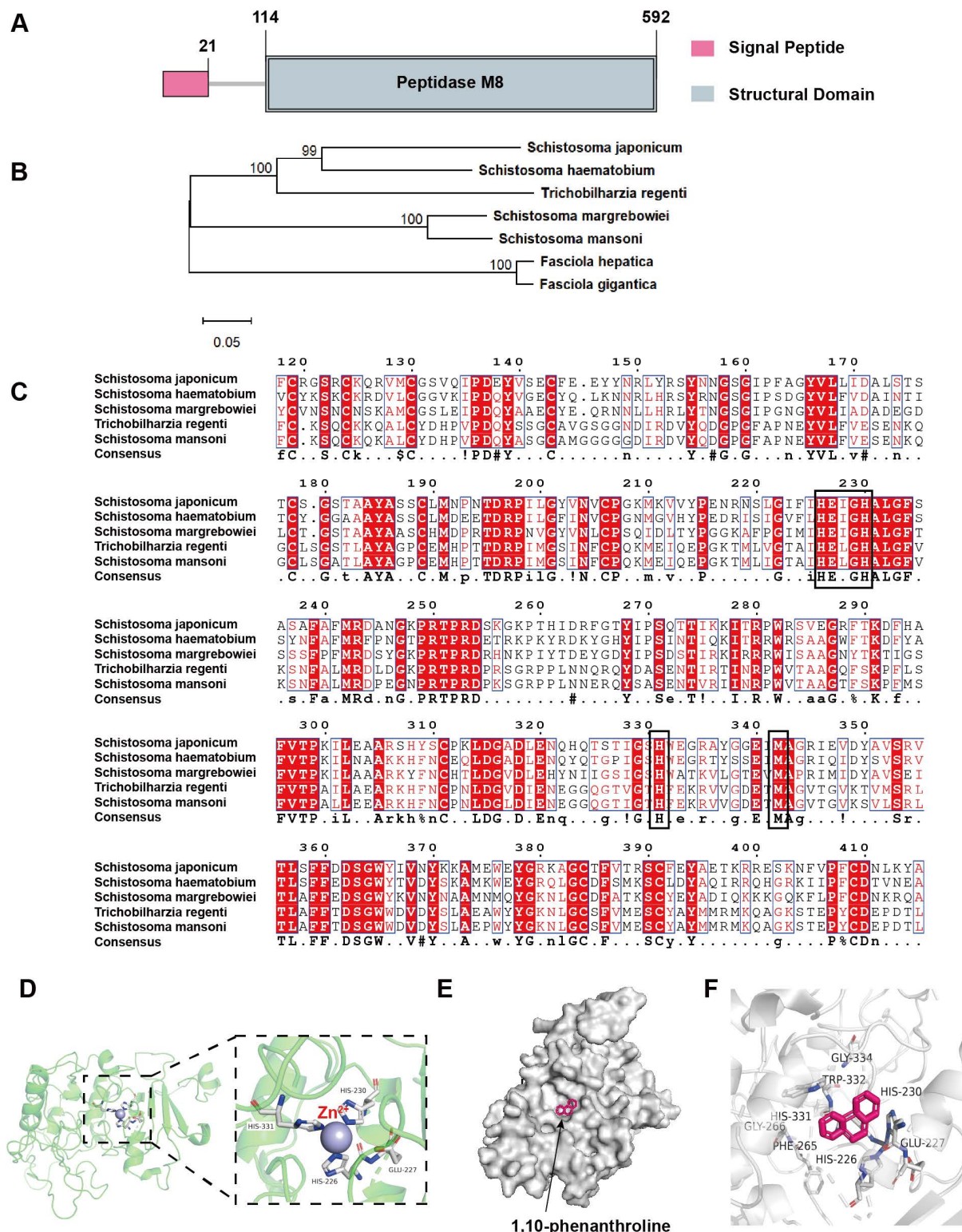

**Fig 1. SjLLPi1 belongs to the M8 matrix metalloprotease family.** (A) Domain annotation of SjLLPi1 was performed using the SMART tool. (B) Phylogenetic tree was constructed using MEGA 11 software. (C) Multiple sequence alignment was conducted by ESPript. Conserved $Zn^{2+}$ binding motif HEXXH and H and M residues were marked with black frames. (D) Homology modeling and (E and F) molecular interaction assay were conducted as described in Materials and Methods.

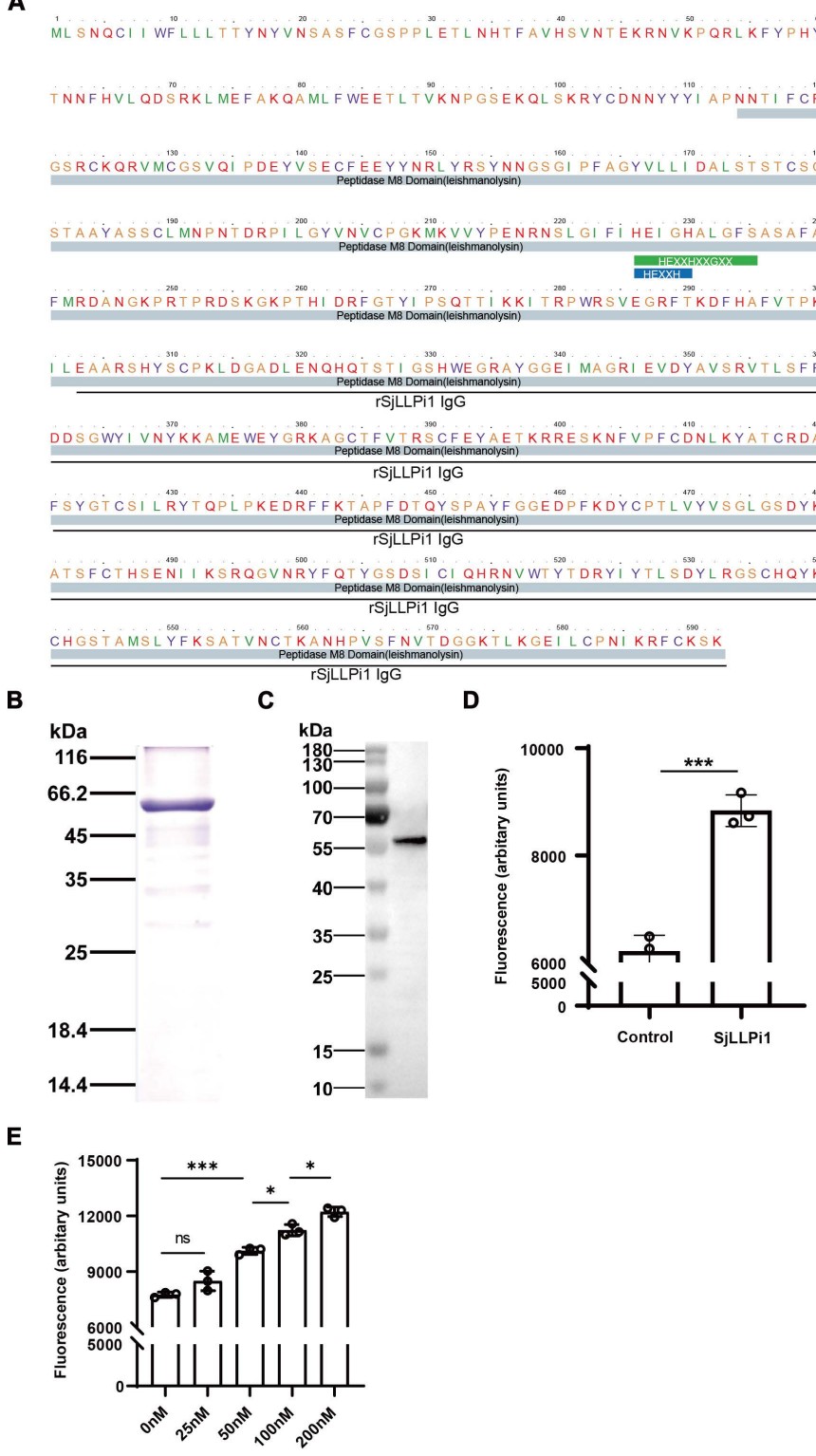

**Fig 2. Expression, purification, and enzymatic activity identification of recombinant SjLLPi1.** (A) SjLLPi1 protein sequence. Peptidase M8 domain was underlined in gray, the sequence employed in the preparation of the anti-SjLLPi1 antibody was underlined in black. (B) Recombinant SjLLPi1 was expressed in *Pichia pastoris*, separated by SDS-PAGE, and stained with Coomassie brilliant blue. (C) Western blotting analysis of the purified SjLLPi1

protein by using anti-SjLLPi1 monoclonal antibody. (D) Enzymatic activity of SjLLPi1 and (E) dose-dependent enzymatic activity of SjLLPi1 was determined by fluorescence analysis as described in Materials and Methods. Data were means±SD of 3 samples from three independent experiments.

Subsequently, we employed fluorescence analysis using Suc-Leu-Tyr-AMC as a fluorescent substrate to assess the hydrolytic enzyme activity of recombinant SjLLPi1, and the results demonstrated a significant enzymatic activity of recombinant SjLLPi1 (Fig 2D). To further validate the enzymatic activity of SjLLPi1, a dose-dependent assay was conducted. The results revealed a significant dose-dependent enzymatic activity of SjLLPi1 (Fig 2E). These findings demonstrate the successful expression and purification of enzymatically active SjLLPi1 protein.

## SjLLPi1 is prominently expressed in cercariae and localized to the acetabular glands and their ducts

To ascertain the expression pattern of SjLLPi1 across different developmental stages of *S. japonicum*, we reanalyzed RNA-seq data from our previous study [34]. The results revealed that SjLLPi1 mRNA levels peak during the sporocyst stage (Fig 3A). Then, results of real-time RT-PCR analysis confirmed the highest expression of SjLLPi1 mRNA during the sporocyst stage (Fig 3B). However, results of western blotting demonstrated that SjLLPi1 protein was undetectable in the egg, miracidium and sporocyst stages, only becoming detectable in the cercarial stage (Fig 3C).

To elucidate the localization of SjLLPi1 protein within the cercariae of *S. japonicum*, we immunofluorescently labeled the SjLLPi1 protein in schistosome cercariae. The results showed that SjLLPi1 protein is predominantly localized in the head region of cercariae, specifically within the acetabular glands and their ducts (Fig 3D). Combining these findings with those in Fig 2D, E suggests that SjLLPi1 may play a role as an enzyme during the invasion of the host skin by *S. japonicum* cercariae, aiding in host penetration.

## Anti-SjLLPi1 antibody treatment reduces worm and egg burden in *S. japonicum* infected mice

To elucidate the role of SjLLPi1 in host invasion by cercariae, we first examined the changes in the hydrolytic enzyme activity of SjLLPi1 following treatment with anti-SjLLPi1 monoclonal antibody. The results indicated a significant decrease in SjLLPi1 hydrolytic enzyme activity after treatment with anti-SjLLPi1 monoclonal antibody when compared to mouse IgG treatment, suggesting that the anti-SjLLPi1 antibody could block or inhibit the hydrolytic function of SjLLPi1 (Fig 4A). To evaluate whether anti-SjLLPi1 antibody also inhibits the enzymatic activity of cercariae during host invasion, we assessed the hydrolytic activity of cercarial secretions and cercarial sonicated supernatant toward the substrate, along with the inhibitory effect of anti-SjLLPi1 antibody. Both cercarial secretions and sonicated supernatant exhibited hydrolytic activity, which was significantly inhibited by anti-SjLLPi1 antibody (Fig 4B, C). Then, we infected mice percutaneously with *S. japonicum* cercariae treated with anti-SjLLPi1 monoclonal antibody or mouse IgG control antibody. After 42 days, the number of adult worms and fecal eggs in the mice were assessed, as depicted in the experimental design in Fig 4D. The results revealed a significant reduction in the number of adult worms in the mice infected with anti-SjLLPi1 antibody treated cercariae (Fig 4E). Furthermore, we observed significant decreases in the number of eggs in the livers (Fig 4F), intestines (Fig 4G), and feces (Fig 4H) of mice from the group treated with anti-SjLLPi1 monoclonal antibody. These findings suggest that treating cercariae with anti-SjLLPi1 antibody leads to reduced numbers of adult worms and eggs within the mice.

## Anti-SjLLPi1 antibody treatment decreases worm burden in mice 7 days after *S. japonicum* infection

Multiple stages are involved in the development of cercariae into adult worms within the definitive host, including skin invasion, migration within the host, and transformation into adult worms in the portal-mesenteric venous system [20]. Therefore, treatment of cercariae with anti-SjLLPi1 antibody resulted less number of adult worms in infected mice, possibly because SjLLPi1 may act on one or more of these developmental stages. The primary migration route of *S. japonicum*

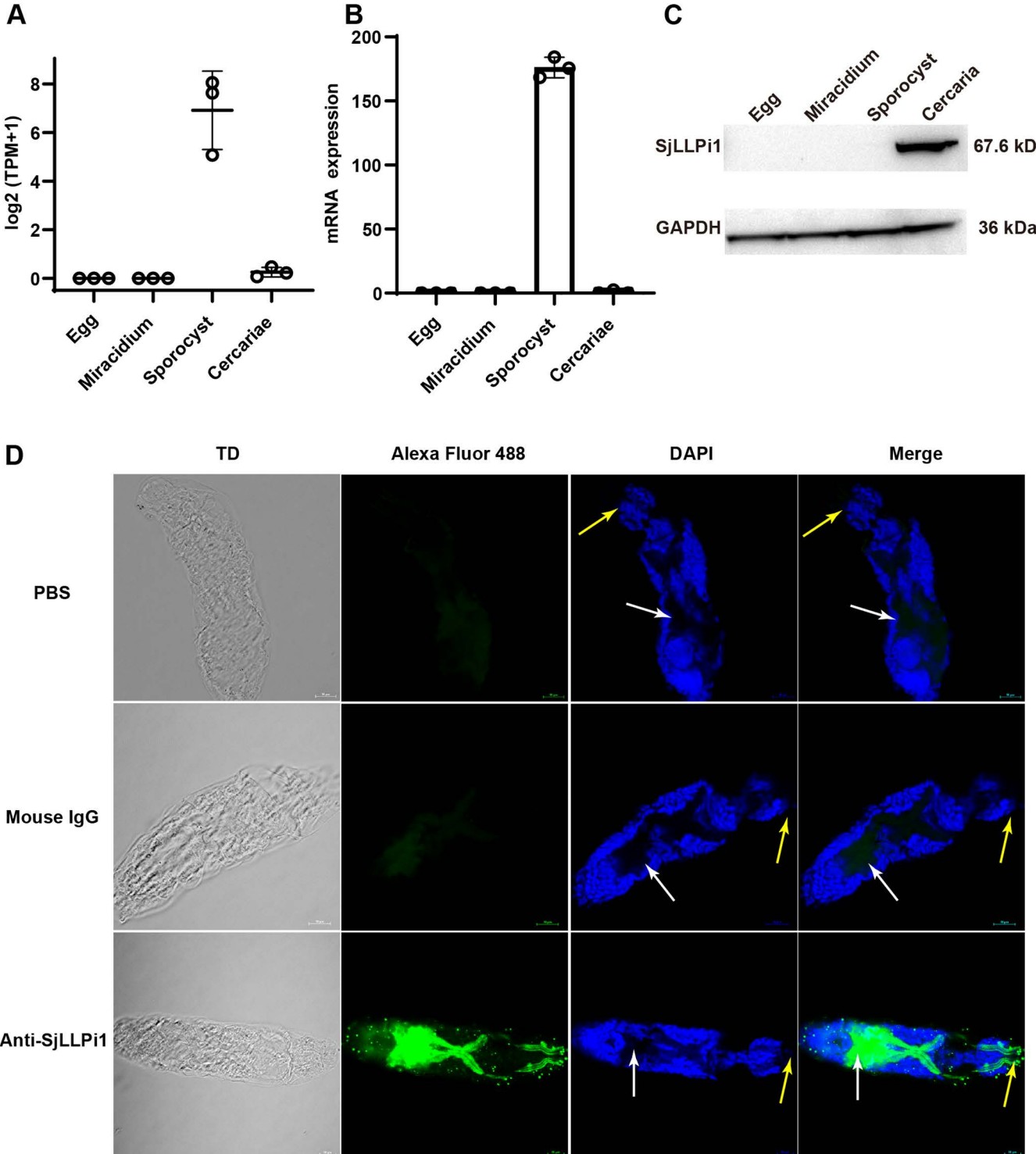

**Fig 3. SjLLPi1 is prominently expressed in cercariae and localized to the acetabular glands and their ducts.** (A) mRNA sequencing data of different developmental stages was obtained from our previous study, and SjLLPi1 mRNA levels were reanalyzed. (B) SjLLPi1 mRNA levels in different developmental stages were detected by real-time RT-PCR. Data were means±SD of 3 samples from three independent experiments. (C) SjLLPi1 protein levels in different developmental stages were detected by Western blotting. (D) Localization of SjLLPi1 protein within the cercariae head was determined by immunofluorescence staining. Yellow arrows indicate anterior region and white arrows indicate acetabular glands.

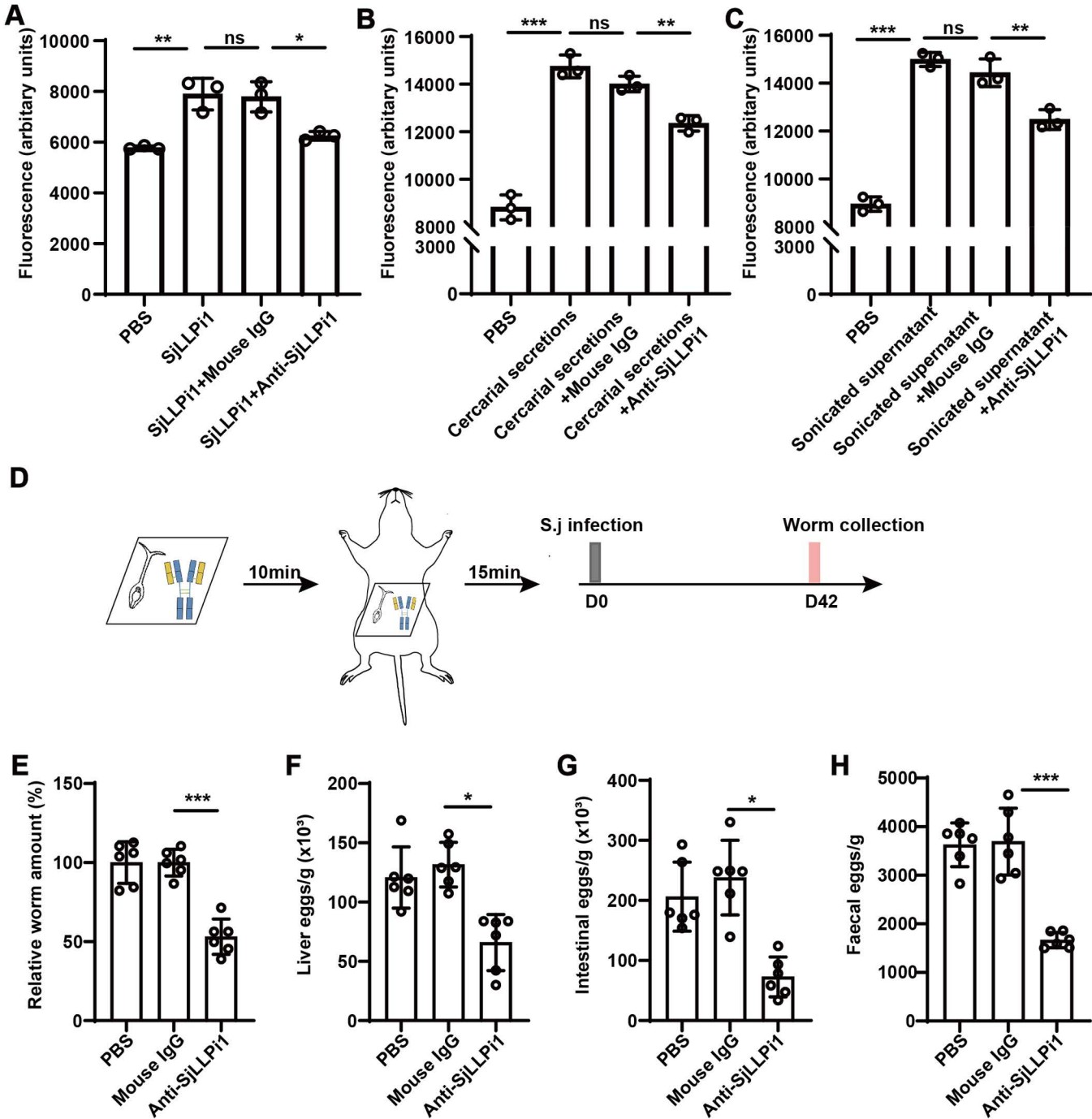

**Fig 4. Anti-SjLLPi1 antibody treatment reduces worm and egg burden in *S. japonicum* infected mice.** (A) Recombinant SjLLPi1 was treated with anti-SjLLPi1 monoclonal antibody or mouse IgG control antibody, enzymatic activity of SjLLPi1 was determined by fluorescence analysis as described in Materials and Methods. Cercarial secretions and cercarial sonicated supernatants were prepared and treated with anti-SjLLPi1 monoclonal antibody or mouse IgG control antibody, enzymatic activities of (B) cercarial secretions and (C) cercarial sonicated supernatants were then assessed as described in the Materials and Methods. Data were means ± SD of 3 samples from three independent experiments. (D) Schematic diagram of the experimental design. 100 *S. japonicum* cercariae were pre-treated with anti-SjLLPi1 monoclonal antibody or mouse IgG control antibody before infecting mice (n = 3 per group). 42 days after infection, the mice were sacrificed. (E) Mice were perfused and adult worms flushed out were collected and counted. (F to H) Egg burden in the liver, intestine, and feces were determined as described in Materials and Methods. Data were means ± SD of 6 mice from two independent experiments.

within the definitive host includes passage from the skin to the lungs and eventually to the portal-mesenteric venous system [20]. 7 days after cercariae enter the definitive host, nearly all parasites have left the lungs [20]. To further investigate which stage or stages of schistosome invasion, migration, or transformation into adult worms SjLLPi1 may target, we infected mice with cercariae treated with anti-SjLLPi1 monoclonal antibody or its control mouse IgG antibody, and then assessed parasite numbers within the mice 7 days later, as outlined in the experimental design in Fig 5A. The results demonstrate a significant decrease in the number of parasites within the mice infected with anti-SjLLPi1 antibody treated cercariae (Fig 5B, C), indicating a substantial reduction in the number of parasites within 7 days after infection. These findings suggest that SjLLPi1 may exert its effects early during cercarial invasion of the skin or parasite migration.

## SjLLPi1 treatment activates BMDM and promotes the production of parasiticidal NO by BMDM

The host immune response plays a critical role in reducing parasite numbers during schistosome migration, with macrophages pivotal in the initial host defense by killing the parasites [35,36]. To explore the potential impact of SjLLPi1 during the early stages of parasite migration, we evaluated its influence on macrophage activation and nitric oxide (NO) production by macrophages, a key parasiticidal factor [35,37]. Mouse BMDM were isolated, and its purity was confirmed at 95.7% using flow cytometry (Fig 6A). Subsequent *in vitro* treatment of BMDM with SjLLPi1 resulted in a significant upregulation of mRNA levels of pro-inflammatory factors TNF-α, iNOS, IL-6, as well as anti-inflammatory factors IL-10 and Arg-1, compared to the untreated control group (Fig 6B, C). Moreover, this effect was observed at all tested concentrations of SjLLPi1 (ranging from 2.5 µg/ml to 40 µg/ml, S1 Fig), indicating substantial activation of BMDM in response to SjLLPi1.

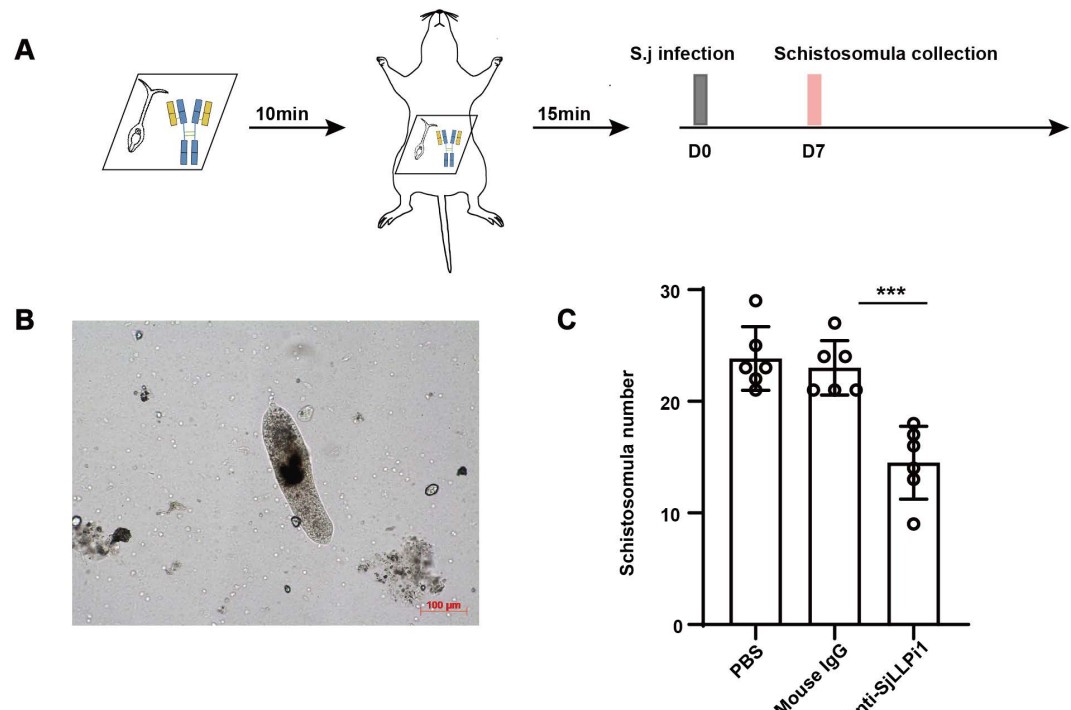

**Fig 5. Anti-SjLLPi1 antibody treatment decreases worm burden in mice 7 days after *S. japonicum* infection.** (A) Schematic diagram of the experimental design. 100 *S. japonicum* cercariae were pre-treated with anti-SjLLPi1 monoclonal antibody or mouse IgG control antibody before infecting mice (n = 3 per group). 7 days after infection, the mice were sacrificed. Mice were perfused and the schistosomula flushed out were collected and counted. (B) Representative morphology of schistosomula. (C) Quantitative analysis of parasite numbers. Data were means ± SD of 6 mice from two independent experiments.

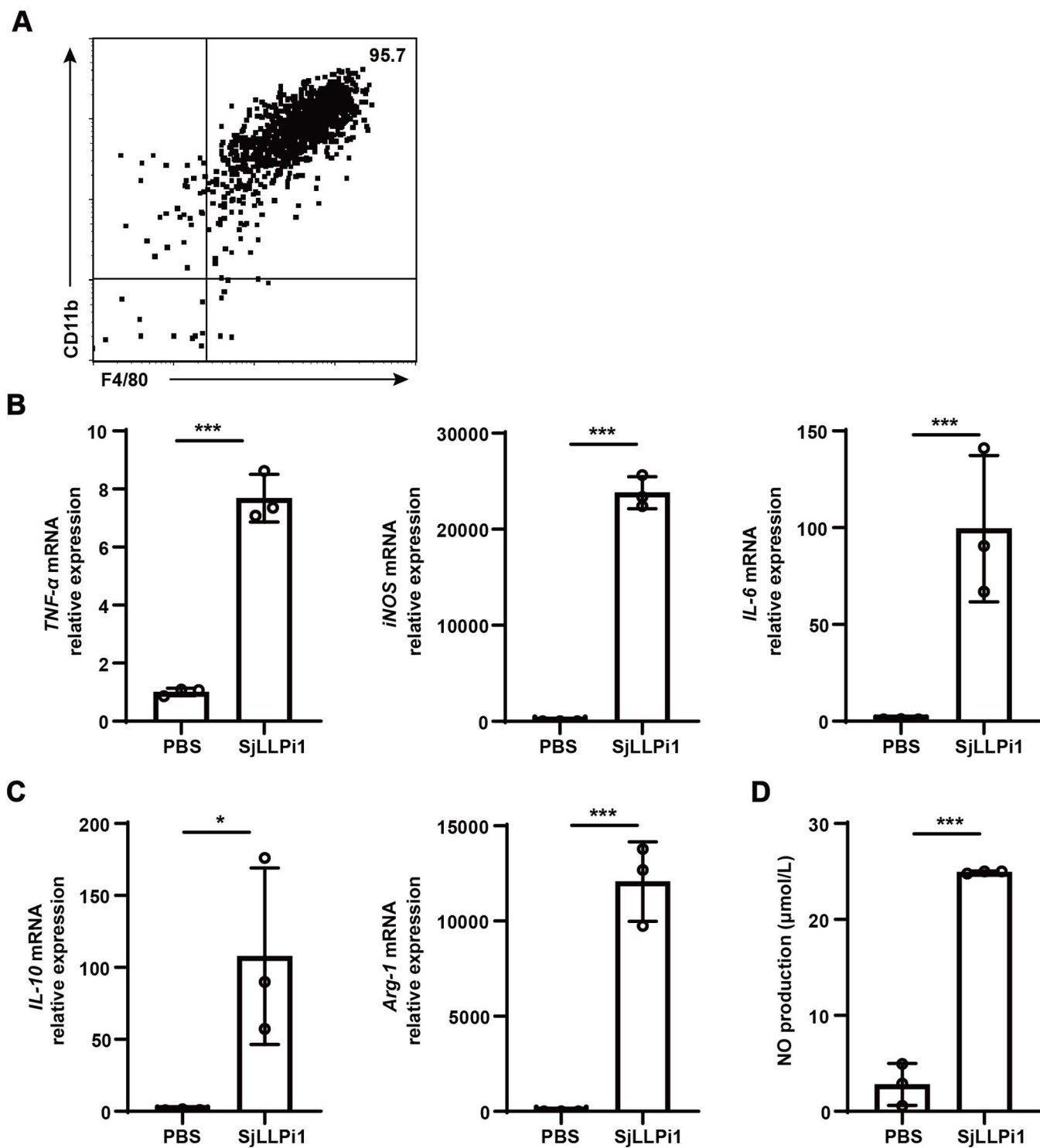

**Fig 6. SjLLPi1 treatment activates BMDM and promotes the production of parasiticidal NO by BMDM.** (A) Mouse BMDM were purified, and the purity of BMDM was determined using flow cytometry. BMDM were isolated from mice and treated with recombinant SjLLPi1, mRNA levels of (B) TNF-α, iNOS, IL-6, as well as (C) IL-10 and Arg-1 were detected by real-time RT-PCR, (D) NO level in the supernatant was determined as described in Materials and Methods. Data were means±SD of 3 samples from three independent experiments.

Direct quantification of NO, a key toxic molecule involved in macrophage-mediated parasite elimination [35,37], revealed a notably higher concentration in the supernatant of SjLLPi1-treated BMDM cultures compared to the control group (Fig 6D). These findings suggest that SjLLPi1 activates macrophage and enhance parasiticidal NO release, implying that SjLLPi1 may not facilitate parasite evasion of host immune attack. In addition, our findings further suggest that the reduction in the number of parasites 7 days after mice infected with anti-SjLLPi1 antibody treated cercariae is likely due to a decrease in the number of cercariae penetrating the skin, rather than an augmented parasiticidal effect of macrophages.

**Anti-SjLLPi1 antibody treatment reduces the number of schistosomula in mice skin 30 min after infection**

To further elucidate whether SjLLPi1 directly influences the number of parasite in the skin stage, following the method described in previous studies [9,38], we infected mice with cercariae treated with anti-SjLLPi1 monoclonal antibody, mice infected with cercariae treated with either mouse IgG or anti-HSP60 monoclonal antibody were used as controls, skin samples from the infection site were collected 30 min later to quantify the number of larvae (Fig 7A). Upon confirming no morphological effects on parasites of the antibodies (S2 Fig), skin-stage parasites were quantified. As illustrated in Fig 7B, parasites that shed their tails and in Fig 7C, parasites that delayed tail shedding were both included in our counts. The results revealed a significant reduction in the number of skin-stage parasite in the skin of mice infected with anti-SjLLPi1 antibody treated cercariae when compared with mouse IgG or anti-HSP60 control antibody treated group (Fig 7D). These results indicate that treatment with anti-SjLLPi1 antibody can decrease the number of skin-stage parasite in infected mice. To investigate the proteolytic activity of SjLLPi1 against proteins from different layers of the skin, we selected representative proteins of skin layers as substrates to assess their susceptibility to hydrolysis by SjLLPi1. Specifically, keratins K1 and K10 were chosen to represent the epidermis, type IV collagen for the basement membrane, and type I collagen for the dermis [39–41]. The results demonstrated that SjLLPi1 showed clear hydrolytic activity against keratin K10 and type I collagen, but not against keratin K1 or type IV collagen (Fig 7E), indicating that the hydrolytic effect of SjLLPi1 on host skin may primarily target the epidermis and dermis. These findings suggest that SjLLPi1 may facilitate cercarial penetration of the host skin.

## Discussion

Schistosomiasis, a neglected tropical disease, ranks as the second most prevalent parasitic infection after malaria [3]. A critical transmission stage of schistosomiasis involves cercariae invading the host's skin, migrating to become adult worms and producing eggs [4]. However, the precise mechanisms governing the invasion of host skin by schistosome cercariae are not fully understood. It is one of crucial issues for skin penetration regarding cercariae release proteases from their acetabular glands to degrade the host's epidermis and dermis by breaking down large molecules [10,13,14]. Current studies on proteases employed by *S. japonicum* cercariae during skin invasion focus on tissue proteases and elastase, which facilitate host skin invasion by hydrolyzing collagen IV, fibronectin, and laminin within the host's skin extracellular matrix [17,26]. Among the proteases produced by *S. japonicum* cercariae, leishmanolysin is the most abundant [21,34,42]. Its role remains mainly explored through omics analyses, leaving its specific functions unclear. In this study, we demonstrated the expression of leishmanolysin in the acetabular glands of cercariae and aiding in skin penetration.

Studies have demonstrated that leishmanolysin functions as a representative enzyme in the M8 matrix metalloproteases, it falls under the metzincins family due to its $Zn^{2+}$-binding motif, HEXXH, which includes two histidine and one glutamate residues acting as $Zn^{2+}$ ligands for catalytic activity [31]. Metzincins proteases share a distinct characteristic of a metal-binding site, typically $Zn^{2+}$, engaging in general protein degradation events during protein digestion and tissue development, as well as specific proteolytic actions that regulate their own activity or that of other enzymes and bioactive peptides [43,44]. Upon elucidating the structure of SjLLPi1, we identified the presence of an N-terminal signal peptide and an M8 matrix metalloprotease domain within the SjLLPi1 sequence. Through aligning leishmanolysin sequences from various schistosome species, conserved sequences within the M8 matrix metalloprotease domain of SjLLPi1 were revealed.

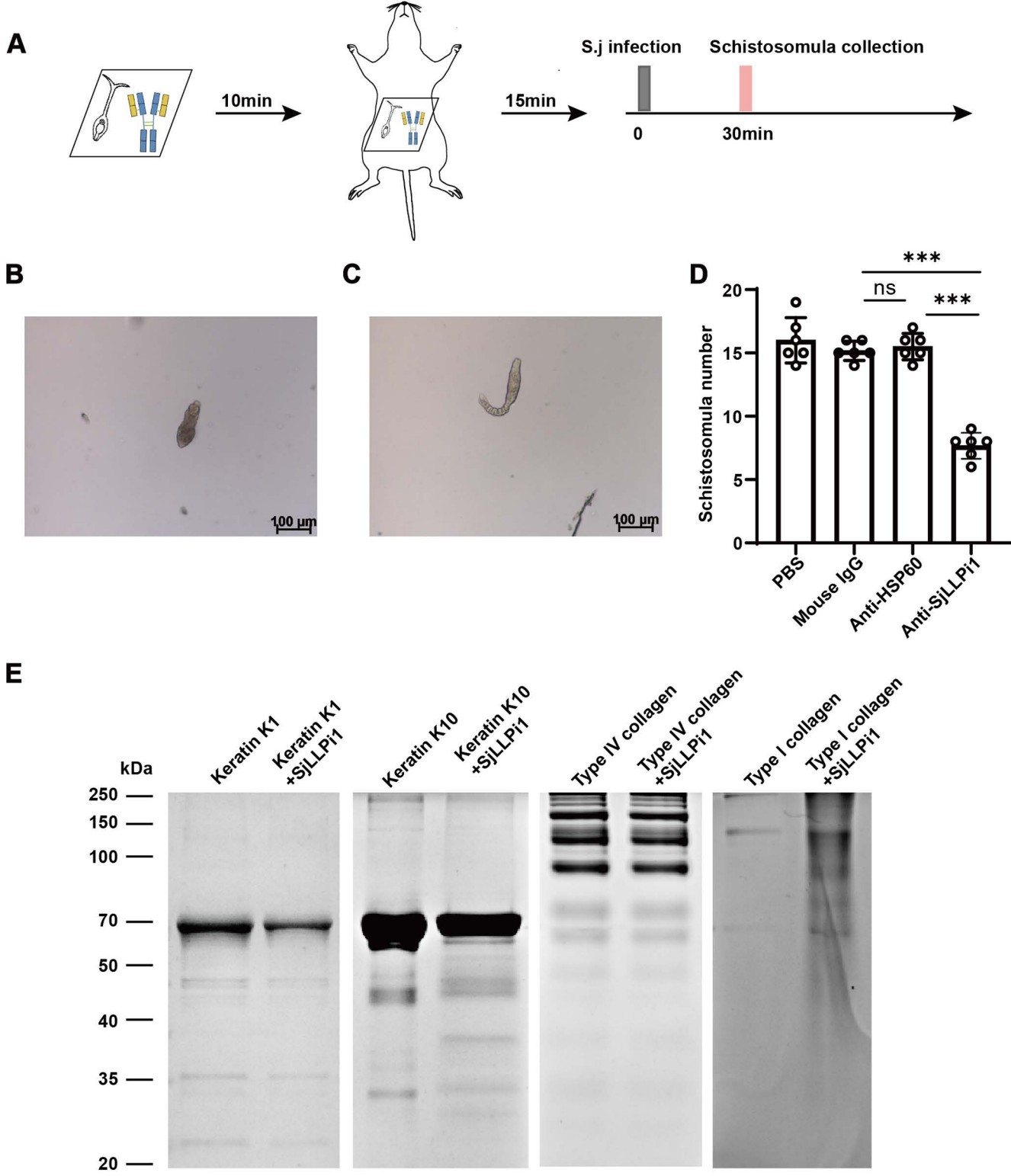

**Fig 7. Anti-SjLLPi1 monoclonal antibody treatment reduces the number of schistosome larvae in the skin of mice 30 min after infection.** (A) Schematic diagram of the experimental design. 50 *S. japonicum* cercariae were pre-treated with anti-SjLLPi1 monoclonal antibody, mouse IgG control antibody or anti-HSP60 monoclonal control antibody before infecting mice (n = 3 per group). 30 min after infection, the mice were sacrificed, the

infected skin was excised, and the number of skin-stage parasite was quantified as described in Materials and Methods. Parasites that shed their tails and parasites that delayed tail shedding were both included in our quantification. (B) Representative morphology of parasites that shed their tails. (C) Representative morphology of parasites that delayed tail shedding. (D) Quantitative analysis of parasite numbers. Data were means ± SD of 6 mice from two independent experiments. (E) The proteolytic activity of SjLLPi1 toward keratins K1 and K10, type IV collagen, and type I collagen was assessed as described in the Materials and Methods.

Consequently, the classification of SjLLPi1 within the M8 matrix metalloprotease family was corroborated through the utilization of homology modeling and molecular interaction analyses. In our investigation of the interaction between SjLLPi1 and $Zn^{2+}$, we ascertained that SjLLPi1 utilizes two histidine and one glutamate residues as $Zn^{2+}$ catalytic ligands, a hallmark of metzincins [31]. As a result, we postulate that SjLLPi1 may exhibit both specific and nonspecific hydrolytic activities resembling those observed in the metzincins family of proteases.

To ascertain the expression and localization of SjLLPi1 in *S. japonicum* cercariae, we first expressed and purified SjLLPi1 in *Pichia pastoris*, confirming its M8 matrix metalloprotease activity. Subsequently, through real-time RT-PCR, we observed significantly higher mRNA levels of SjLLPi1 in the sporocyst stage compared to the cercaria stage. This finding is consistent with previous study that detected elevated expression of leishmanolysin in the sporocyst stage compared to the cercariae stage in *S. mansoni* [45]. Previous study on *S. mansoni* used incorporation of the cytosine analog 5-ethynyl-uridine (EU) and fluorescence labeling to detect active transcription in the cercariae stage, the results showed that fluorescence signals were detected in less than 10% of the cercariae, indicated there is very little transcription in cercariae [46]. This may explain the notable decrease in mRNA levels of SjLLPi1 in cercariae than that in sporocyst. Previous study on *S. mansoni* have shown that the leishmanolysin is also expressed, detected by both quantitative RT-PCR and western blotting, at the early miracidia stage [47], suggesting a role in the invasion of the intermediate snail host. However, in our study, SjLLPi1 mRNA was primarily detected at the sporocyst stage by RNA-seq and quantitative RT-PCR, whereas the SjLLPi1 protein could only be detected in the cercarial stage of *S. japonicum* by western blotting. This indicates that the leishmanolysin of *S. japonicum* may differ from that of *S. mansoni*, with its protein expression predominantly in the cercarial stage, potentially contributing to the invasion of the definitive mammal hosts. Immunofluorescence analysis showed that SjLLPi1 is predominantly expressed in the acetabular glands and gland ducts of cercariae, suggesting that this protease may be synthesized in the acetabular glands, released through the gland ducts, and ultimately involved in hydrolytic activities during the invasion of the host's skin.

We then demonstrated the role of SjLLPi1 in the invasion of the definitive host by cercariae. After confirming that anti-SjLLPi1 monoclonal antibody significantly inhibit the enzymatic activity of SjLLPi1, we treated cercariae with the antibody and infected mice with the treated cercariae. 42 days post-infection, we observed a significant reduction in the number of adult worms and egg burden in the mice. Considering the various stages within the host, from cercariae penetration (can occur within minutes) to migration (migrate through the lung and left the lung 7 days post infection) and transformation into adult worms in the portal-mesenteric veins (21 days post infection) [12,20,48], we further examined the number of parasites migrating from the skin to the portal vein in mice infected with cercariae treated with anti-SjLLPi1 antibody 7 days after infection. The results showed a significant decrease in the number of parasites that collected from the portal vein 7 days after infection, suggesting that SjLLPi1 may play a role in the early stages of cercariae invasion of the skin or parasite migration. However, the specific mechanism, whether through its own hydrolytic activity aiding in skin penetration by cercariae or by protecting migrating parasites from immune attack, requires further elucidation.

Macrophages are essential innate immune cells pivotal in defense against parasites [35,36]. When activated, they exert parasiticidal effects during the invasion of cercariae and the migration of the parasites, primarily through nitric oxide (NO) [35,37]. Studies reveal that schistosome cercariae secrete proteins hindering host macrophage activation. For instance,

Sm16 from *S. mansoni* interferes with TLR3 and TLR4 signaling, reducing IL-12, IL-10, and IFN-γ production, and impeding antigen presentation of macrophage [49]. Leishmanolysins from other parasites also modulate macrophage activation. For example, leishmanolysin GP63 from leishmania cleaves various molecules through hydrolysis, such as protein tyrosine phosphatases, c-Jun, mTOR, or NLRP3 inflammasome components, thereby inhibiting signal transduction and pro-inflammatory function of host macrophages, resulting in decreased levels of NO, TNF-α, IL-12, ROS, and ultimately promoting parasite survival [50–53]. Similarly, leishmanolysin from *Trypanosoma cruzi* can inhibit NO production in host macrophages, thereby attenuating the inflammatory response initiated by macrophages [54]. However, the role of SjLLPi1 in protecting schistosomes against parasiticidal effects of host macrophages remains unclear. Our data showed that treatment of macrophages with SjLLPi1 upregulates both pro-inflammatory (TNF-α, iNOS, IL-6) and anti-inflammatory (IL-10, Arg-1) molecules, including NO, suggesting SjLLPi1 does not protect schistosomes from macrophage-induced parasiticidal effects by inhibiting macrophage activation. Therefore, we hypothesize that SjLLPi1 may act as a hydrolase, aiding the penetration of cercariae into the host skin during invasion.

Upon infecting mice with cercariae treated with anti-SjLLPi1 monoclonal antibody, a significant reduction in the number of parasites in the mouse skin was observed 30 min post-infection. Although anti-SjLLPi1 antibody significantly reduced number of skin-stage larvae, it did not completely prevent cercarial invasion. This partial inhibition may result from incomplete neutralization of protease activity during dynamic process of protease secretion and host skin degradation. In addition, the secretion of multiple proteases by cercariae, such as serine, cysteine, and aspartic types [21], suggests functional redundancy, whereby inhibition of a single enzyme is insufficient due to compensatory mechanisms. Such redundancy likely represents an evolutionary strategy to ensure successful host entry under varying environmental or immune conditions. Our results suggest that SjLLPi1 facilitates the invasion of cercariae into the host skin. Due to the technological limitations, the detailed mechanism by which SjLLPi1 exerts its hydrolytic action during cercarial invasion of the host is difficult to be verified at current stage. However, existing data strongly indicate that SjLLPi1 likely aids cercariae in invading the host by hydrolyzing host skin components.

In conclusion, this study for the first time revealed that the highly expressed SjLLPi1 in cercariae acetabular glands and their ducts assists the penetration of cercariae into the skin by its enzymatic hydrolysis. This research provides insights into the role of SjLLPi1 in schistosome invasion and may inform future strategies aimed at interfering with early stages of host entry.

## Supporting information

**S1 Data. Materials and methods and figure legends relevant to the supporting figures.**
(DOC)

**S2 Data. Raw source data.** Underlying data used for graph generation, as well as full-size original gel images, unedited blot images, and raw microscopy data.
(XLSX)

**S1 Fig. Macrophage activation induced by different concentrations of SjLLPi1.** BMDM were isolated from mice and treated with different concentrations recombinant SjLLPi1, mRNA levels of (A) TNF-α, (B) iNOS, (C)IL-6, as well as (D) IL-10 and (E) Arg-1 were detected by real-time RT-PCR. Data were means ± SD of 3 samples from three independent experiments.
(TIF)

**S2 Fig. Morphology examination of schistosome larvae by carmine staining.** (A) Cercariae were incubated with anti-SjLLPi1 antibody or mouse IgG or anti-HSP60 antibody for 10 min, morphology of cercariae was examined by carmine staining as described in the Materials and Methods (Scale bar, 50 μm). (B) Mice were infected with anti-SjLLPi1 antibody

or mouse IgG or anti-HSP60 antibody treated cercariae, schistosomula were collected from the skin 30 min post-infection, morphology of schistosomula was examined by carmine staining as described in the Materials and Methods (Scale bar, 25 μm).
(TIF)

## Author contributions

**Conceptualization:** Jifeng Zhu, Wei Hu, Chuan Su.

**Data curation:** Fanyan Chen, Bingkuan Zhu, Yu Fang, Jifeng Zhu.

**Funding acquisition:** Chuan Su.

**Investigation:** Fanyan Chen, Bingkuan Zhu, Yu Fang, Zilüe Li, Zhigang Lei, Zechao Xue, Tao Shen, Sha Zhou, Xiaojun Chen, Lei Xu, Yalin Li.

**Methodology:** Fanyan Chen, Bingkuan Zhu, Yu Fang, Jifeng Zhu.

**Project administration:** Jifeng Zhu, Chuan Su.

**Software:** Fanyan Chen, Bingkuan Zhu, Jifeng Zhu.

**Writing – original draft:** Zilüe Li, Zhigang Lei, Jifeng Zhu.

**Writing – review & editing:** Jifeng Zhu, Wei Hu, Chuan Su.

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
