## [Decision Letter · Decision Letter 0]

19 Apr 2025

PPATHOGENS-D-25-00386

Schistosoma japonicum leishmanolysin SjLLPi1 facilitates the invasion of cercariae into the host skin

PLOS Pathogens

Dear Dr. Su

Thank you for submitting your manuscript to PLOS Pathogens. After careful consideration, we feel that it has merit but does not fully meet PLOS Pathogens's publication criteria as it currently stands. Therefore, we invite you to submit a revised version of the manuscript that addresses the points raised during the review process. Reviewer two has outlined a number of substantial issues to address.

Please submit your revised manuscript within 60 days (by 11th June). If you will need more time than this to complete your revisions, please reply to this message or contact the journal office at plospathogens@plos.org. Please include the following items when submitting your revised manuscript:

We look forward to receiving your revised manuscript.

Kind regards,

Paul Giacomin

Academic Editor

PLOS Pathogens

Jeffrey Dvorin

Section Editor

PLOS Pathogens

 Sumita Bhaduri-McIntosh

Editor-in-Chief

PLOS Pathogens

orcid.org/0000-0003-2946-9497

 Michael Malim

Editor-in-Chief

PLOS Pathogens

orcid.org/0000-0002-7699-2064

**Journal Requirements:**

At this stage, the following Authors/Authors require contributions: Fanyan Chen, Bingkuan Zhu, Zilüe Li, Zhigang Lei, Zechao Xue, Tao Shen, Sha Zhou, Xiaojun Chen, Yalin Li, Jifeng Zhu, Wei Hu, and Chuan Su. Please ensure that the full contributions of each author are acknowledged in the "Add/Edit/Remove Authors" section of our submission form.

2) We noticed that you used the phrase 'data not shown' in the manuscript. We do not allow these references, as the PLOS data access policy requires that all data be either published with the manuscript or made available in a publicly accessible database. Please amend the supplementary material to include the referenced data or remove the references.

- ® on page: 11.

5) Tables should not be uploaded as individual files. Please remove these files and include the Tables in your manuscript file as editable, cell-based objects. For more information about how to format tables, see our guidelines:

https://journals.plos.org/plospathogens/s/tables 

6) We note that your Data Availability Statement is currently as follows: "The authors confirm that the data supporting the findings of this study are available within the article and its Supplementary material.". Please confirm at this time whether or not your submission contains all raw data required to replicate the results of your study. Authors must share the “minimal data set” for their submission. PLOS defines the minimal data set to consist of the data required to replicate all study findings reported in the article, as well as related metadata and methods (https://journals.plos.org/plosone/s/data-availability#loc-minimal-data-set-definition).

- The points extracted from images for analysis..

7) Please amend your detailed Financial Disclosure statement. This is published with the article. It must therefore be completed in full sentences and contain the exact wording you wish to be published. Please ensure that the funders and grant numbers match between the Financial Disclosure field and the Funding Information tab in your submission form. Note that the funders must be provided in the same order in both places as well.

**Reviewers' Comments:**

Reviewer's Responses to Questions

**Part I - Summary**

Reviewer #1: The novelty, significance, planning and general execution are remarkable. The weakness are due to the fact that the important information generated may not be translated into a drug or vaccine for the dreadful disease.

Reviewer #2: Chen et al. have generated an antibody against the SjLLPi1and show that preincubation with the antibody targeting the leishmanolysin reduces the enzymatic activity of the protein and that the infection is reduced in a mouse model 42 and 7 days post infection. Moreover, they show that a lower number of schistosomula are identified in the skin 30 minutes post-infection, suggesting that the leishmanolysin may play a role in the penetration of the skin.

Reviewer #3: This manuscript describes the initial characterization of Schistosoma japonicum leishmanolysin SjLLPi1. This is novel because while leishmanilysin enzymes have been described in S. mansoni, they have not been described in S. japonicum. These authors successfully identified SjLLPi1, expressed it as a recombinant protein, and generated monoclonal antibodies which were used to localize it to S. japonicum cercariae heads in the acetabular glands as well as demonstrate it's role in skin penetration in a mouse model. Some weaknesses include details that need to be clarified in the materials and methods. Overall, this is a well constructed manuscript describing a key enzyme in S. japonicum facilitating infection, and a potential therapeutic target.

**Part II – Major Issues: Key Experiments Required for Acceptance**

Reviewer #1: None.

Reviewer #2: 1. The authors have identified and expressed S. japonicum leishmanolysin SjLLPi1 and identified this as a protein expressed in cercariae only.

- Previous studies (Hambrook et al PPAT 2018) using S. mansoni indicate that the protein is expressed also early in miracidia, applying a role in the invasion of the snail. The authors do not refer to this paper, which should be considered. Also, a more detailed kinetics study could be considered to claim that the SjLLPi1 is only in the cercariae.

2. A dose dependent activity of the SJLLPi should be shown in figure 2

3. The enzymatic activity was tested in vitro using a recombinant enzyme. However, it is not clear whether the effect of the antibody is on the inhibition of the enzyme or if the coating of the cercariae with antibodies directed against the parasite per se increases the killing of the worm.

- Ideally, the expression of the protein would be inhibited using molecular methods to avoid coating the parasites with antibodies, which may effectively trigger the immune system. Previous data suggest that sporocysts can be targeted by siRNAs (Hambrook, 2018).

- Alternatively, can the enzymatic effect of the cercariae be tested to show the impact of antibody treatment on degradation, as shown in Figure 2D?

4. As stated above, coating the parasite with antibodies could affect the immediate immune response in the skin. If the SjLLPi1 cannot be targeted by molecular methods, a control for the effect of specific antibodies would be appropriate. For example, could the antibody to SjLLPi1 be compared to another antibody that coats the cercariae but does not inhibit the enzyme? Alternatively, could the effect of IgG from an immunized mouse on cercariae be tested 30 minutes after infection? In both cases, it would be relevant to determine the effect of the coating and compare it with the effects of antibodies generated against SjLLPi1.

5. Importantly, the cercariae must penetrate the skin which likely would involve degradation of keratin. Can the effects of SjLLPi1 on keratin, be measured? If so, how would the SjLLPi1 compare to elastases previously described in S. japonicum? Such data would support the role of the enzyme in skin penetration. Alternatively, which layer of the skin would be targeted by SjLLPi1 during penetration?

Reviewer #3: No new experiments needed for this manuscript.

**Part III – Minor Issues: Editorial and Data Presentation Modifications**

Reviewer #1: 1- Immunofluorescence staining was not described in the Materials and Methods section and may not be replaced by the sentence" we immunofluorescently labeled the protein in schistosome cercariae". Please, use arrows to mark the anterior region and acetabular glands in Figure 3C.

2- In the Materials and Methods, Collection of parasites section should be placed before Mice infection.

3- Last section of the Results is misplaced and must be placed in the Results and Discussion sections before "antibody treatment reduces worm and egg burden in S. japonicum-infected mice".

4- The authors are kindly requested to discuss, generalize, and/or explain this sharp discrepancy between mRNA production and protein expression in both the sporocyst and cercarial stages.

5- Please discuss the limitations of the experiments using antibody-coated cercariae regarding antibody inhibitory activity and redundancy of proteases-mediated cercarial infection, based on the significant but limited effect of antibody coating of cercariae before skin invasion.

6- Caution should be shown in the last sentence of the text as it is nearly impossible for the host to elicit memory immune responses (The privilege of vaccines) to the leishmanolysin at few to 48 hours after exposure to cercariae. Drugs also are not expected to be considered.

Reviewer #2: In figure 6, the effects of the recombinant SjLLPi1 on macrophages are shown. Are the levels of SjLLPi1 used when stimulating macrophages biologically relevant levels? Please comment

What the representative morphology reflects should better indicated in figure 7 or the legend. Also, it would be informative to know if the antibody treatment affected the morphology or if that was similar in both groups.

It is more relevant to show the SD instead of SEM as the variation within the group is the concern. Preferably individual data (dots) rather than bars are shown.

Typos noted:

Leishmania major line 4, p 9 should be in italic.

Spelling line 15 p 15: Should be Phylogenetic.

Line 19 p 36 figure (7) text last C should be D.

Reviewer #3: Several items should be addressed in the materials and methods to improve clarity of the manuscript, as listed below:

Materials and methods

1. Please briefly describe how S. japonicum cercaria were collected from Oncomelania hupensis infected with S. japonicum, and how the cercariae were prepared before infecting mice.

2. Please describe the number of mice used per group for infection studies.

3. Please briefly describe the percutaneous mouse infection procedure with S. japonicum, or provide a citation that gives more details of the infection procedure. Were mice anesthetized during the infection procedure? If so, what anesthetic was used? Was the skin depilated and cleaned prior to percutaneous infection?

4. For infection of mice with the SjLLPi1 antibody treated S. japonicum cercariae, how much SjLLPi1 or control mouse IgG antibody was used to pre-treat the cercariae? Were the cercariae washed after treatment with antibody to remove unbound antibody before infecting mice?

5. For quantification of liver and intestinal eggs, please include a statement that mice were humanely euthanized, then tissues were collected post mortem. Also, at what time point after infection were tissues collected for egg quantification?

6. Please describe the methods used for counting eggs.

7. For the schistosomula collection, what buffer was used to perfuse mice to collect schistosomula?

8. For generation of BMDM, how were red blood cells removed?

9. How many replicates of BMDM were used for treatment with SjLLpi1 or controls?

10. Please include a description of the immunophenotyping procedure on BMDM in the materials and methods section.

Results

1. In page 19, line 15, please correct the spelling of “Phylogenetic”

PLOS authors have the option to publish the peer review history of their article (what does this mean? ). If published, this will include your full peer review and any attached files.

**Do you want your identity to be public for this peer review?** For information about this choice, including consent withdrawal, please see our Privacy Policy .

Reviewer #1: **Yes: ** Rashika El Ridi

Reviewer #2: No

Reviewer #3: **Yes: ** Kathryn M. Jones

**Figure resubmission:**
---

## [Decision Letter · Decision Letter 1]

12 Aug 2025

Dear Prof. Su,

We are pleased to inform you that your manuscript 'Schistosoma japonicum leishmanolysin SjLLPi1 facilitates the invasion of cercariae into the host skin' has been provisionally accepted for publication in PLOS Pathogens.

Best regards,

Paul Giacomin

Academic Editor

PLOS Pathogens

Jeffrey Dvorin

Section Editor

PLOS Pathogens

Sumita Bhaduri-McIntosh

Editor-in-Chief

PLOS Pathogens

orcid.org/0000-0003-2946-9497

Michael Malim

Editor-in-Chief

PLOS Pathogens

orcid.org/0000-0002-7699-2064

Reviewer Comments (if any, and for reference):

Reviewer's Responses to Questions

**Part I - Summary**

Reviewer #1: The manuscript is perfect now.

Reviewer #2: See 1st review

**Part II – Major Issues: Key Experiments Required for Acceptance**

Reviewer #1: None.

Reviewer #2: The authors have responded to my concerns and have include the requested text and additional experiments. The new data support their claims and have improved the manuscript.

**Part III – Minor Issues: Editorial and Data Presentation Modifications**

Reviewer #1: All issues have been taken into the closest consideration, except for the PBS used to perfuse mice. An anti coagulant should be included. We routinely used sterile (autoclaved) 7.5% sodium citrate/8.5% NaCl.

Reviewer #2: The gating strategy/process for figure 6A should be shown.

PLOS authors have the option to publish the peer review history of their article (what does this mean? ). If published, this will include your full peer review and any attached files.

**Do you want your identity to be public for this peer review?** For information about this choice, including consent withdrawal, please see our Privacy Policy .

Reviewer #1: **Yes: ** Rashika El Ridi

Reviewer #2: No

---

## [Editor Report · Acceptance letter]

Dear Prof. Su,

We are delighted to inform you that your manuscript, "Schistosoma japonicum leishmanolysin SjLLPi1 facilitates the invasion of cercariae into the host skin," has been formally accepted for publication in PLOS Pathogens.

Best regards,

Sumita Bhaduri-McIntosh

Editor-in-Chief

PLOS Pathogens

orcid.org/0000-0003-2946-9497

Michael Malim

Editor-in-Chief

PLOS Pathogens

orcid.org/0000-0002-7699-2064